# Understanding the Mechanisms and Treatment of Heart Failure: Quantitative Systems Pharmacology Models with a Focus on SGLT2 Inhibitors and Sex-Specific Differences

**DOI:** 10.3390/pharmaceutics15031002

**Published:** 2023-03-20

**Authors:** Jean François Ndiaye, Fahima Nekka, Morgan Craig

**Affiliations:** 1Department of Mathematics and Statistics, Université de Montréal, Montréal, QC H3C 3J7, Canada; 2Sainte-Justine University Hospital Research Centre, Montréal, QC H3T 1C5, Canada; 3Faculty of Pharmacy, Université de Montréal, Montréal, QC H3C 3J7, Canada

**Keywords:** heart failure, compensatory mechanisms, treatments, SGLT2, mathematical modeling, cardiorenal model, fluid and solutes homeostasis, sex-specific, sex-based therapies

## Abstract

Heart failure (HF), which is a major clinical and public health challenge, commonly develops when the myocardial muscle is unable to pump an adequate amount of blood at typical cardiac pressures to fulfill the body’s metabolic needs, and compensatory mechanisms are compromised or fail to adjust. Treatments consist of targeting the maladaptive response of the neurohormonal system, thereby decreasing symptoms by relieving congestion. Sodium–glucose co-transporter 2 (SGLT2) inhibitors, which are a recent antihyperglycemic drug, have been found to significantly improve HF complications and mortality. They act through many pleiotropic effects, and show better improvements compared to others existing pharmacological therapies. Mathematical modeling is a tool used to describe the pathophysiological processes of the disease, quantify clinically relevant outcomes in response to therapies, and provide a predictive framework to improve therapeutic scheduling and strategies. In this review, we describe the pathophysiology of HF, its treatment, and how an integrated mathematical model of the cardiorenal system was built to capture body fluid and solute homeostasis. We also provide insights into sex-specific differences between males and females, thereby encouraging the development of more effective sex-based therapies in the case of heart failure.

## 1. Introduction

Heart failure syndrome was first designated as an emerging epidemic in 1997 and remains a major clinical and public health problem [1]. Affecting more than 64 million people worldwide, heart failure is associated with considerable morbidity and mortality, limited functional capacity, reduced quality of life, and substantial financial burden [2]. It remains a highly prevalent disease among older adults and is associated with a significant risk of mortality within the first year after diagnosis. HF is defined as a complex clinical condition in which the heart is unable to meet metabolic requirements and accommodate venous return due to inadequate cardiac output (CO) [3]. Patients with HF usually exhibit symptoms like dyspnea, fatigue, reduced exercise tolerance, and pulmonary and peripheral edema [4]. HF is caused by structural abnormalities of the heart, functional abnormalities, and other triggering factors [5]. The most common etiologies are ischemic heart disease, hypertension, and diabetes [3,4,5]. Other causes include cardiomyopathies, valvular disease, uncontrolled arrythmias, myocarditis, pericarditis, and toxins (e.g., alcohol, cytotoxic drugs).

When CO is reduced, and there is insufficient tissue perfusion, the body’s neurohormonal mechanisms, such as the renin angiotensin aldosterone system (RAAS) [6], are activated. This activation results in the accumulation of excessive volume overtime, thus increasing both the heart’s preload and afterload, which causes detrimental cardiac remodeling. Furthermore, excessive preload raises venous pressure leading to the onset of peripheral edema and pulmonary congestion and ultimately contributing to hospitalization for HF [6,7]. Furthermore, HF is very common in patients with chronic kidney disease (CKD), and, if not controlled, it is associated with a rapid deterioration of renal function [8].

Many HF treatments have been developed over the years. Most of them are anti-hypertensive therapies and aim to counteract the deleterious effects of the RAAS in the case of heart failure [9,10,11]. Others, such as diuretics, are also very important and serve to relieve congestion [12].

Sodium–glucose co-transporter 2 (SGLT2) inhibitors, which are a newer class of drugs primarily used to treat diabetic patients by inhibiting the glucose reabsorption in the proximal tubule of the kidney, were unexpectedly found to significantly reduce HF hospitalization and cardiovascular death in patients with HF having a reduced ejection fraction (HF-rEF) [13,14]. However, the mechanisms through which these inhibitors act to provide these benefits are incompletely understood.

Quantitative system pharmacology (QSP) is an emerging approach in pharmacological research that facilitates an improved understanding of drug–organ interaction mechanisms and enables the prediction of pharmacological outcomes and treatment improvements. QSP models can integrate a broad variety of data types, including those arising from preclinical and clinical experiments, to establish the multitude of interactions between xenobiotics and physiological systems, such as the heart or kidney. The goal of this approach is to comprehend the overall behavior of these complex systems in a quantitative and predictable way [15]. QSP models are also used to optimize treatments by relating the quantity of drug dosed, frequency of dosing, pharmacokinetics (PK), and pharmacodynamics (PD) with biological or physiological responses [16].

In this review, we first aim to describe the pathophysiology of HF, existing therapies, and the pleiotropic effects of SGLT2 inhibitors that are thought to induce all the unexpected benefits in HF patients. We then describe key elements of existing cardiorenal models, and the role of QSP modeling in clinical decision making by providing a predictive framework to improve therapeutic scheduling and strategies, including preclinical drug development. We also discuss how sex-specific differences could be important to consider for better management of HF by encouraging the development of sex-based treatments.

## 2. Pathophysiology of Heart Failure

Heart failure can be categorized based on various factors such as the circulatory system that is affected (right-sided or left-sided), the type of cardiac function that is compromised (systolic or diastolic), or the underlying pathophysiological factor (pressure-induced or volume-induced) [17]. For example, patients with left ventricular dysfunction may present either systolic dysfunction (i.e., impaired ventricular contraction and ejection) or diastolic dysfunction, which refers to impaired relaxation and ventricular filling. The type of dysfunction that an HF patient has (whether systolic or diastolic) is determined by their ejection fraction, which is defined as the amount of blood pumped out from the ventricle in one heartbeat [3]. Systolic dysfunction occurs when the ejection fraction is less than 40%, thereby creating a condition called heart failure with a reduced ejection fraction (HF-rEF). If the ejection fraction is more than 50%, it is classified as diastolic dysfunction (also called heart failure with a preserved ejection fraction (HF-pEF)). Conversely, an ejection fraction between 40% and 50% is called heart failure with a mid-range ejection fraction (HF-mrEF). The latter may consist of mixed left ventricular dysfunction, meaning that there is a combination of systolic and diastolic dysfunction. It is essential to understand the pathophysiological mechanisms that lead to heart failure for a better management of the disease and to initiate adequate therapeutic options. Some important causes of HF and their pathologic mechanisms are known, and we briefly review these in the sections below.

### 2.1. Pathophysiological Differences between HF-pEF and HF-rEF

Patients with HF-pEF are more likely to be older, female, obese, and have a history of hypertension and/or atrial fibrillation and type 2 diabetes mellitus. Currently, there is unfortunately no currently approved therapy to improve these patients’ outcomes [18]. HF-rEF is typically caused by conditions such as ischemic heart disease (i.e., myocardial infarction), problems with heart valves (e.g., aortic stenosis, mitral regurgitation), or uncontrolled high blood pressure. Patients with HF-rEF can be treated more effectively through the use of medication, surgery, or intervention.

The primary structural alteration in HF-rEF involves eccentric remodeling, which is characterized by chamber dilation that ultimately causes a volume overload. This leads to forward failure as a consequence of a significant anterior myocardial infarction [17,18]. In contrast to HF-rEF, HF-pEF is associated with deficient ventricular relaxation and filling, along with increased ventricular stiffness, which leads to elevated filling pressure due to pressure overload. The alteration is marked by a concentric remodeling or ventricular hypertrophy, which results in pressure overload and predominantly causes backward heart failure.

### 2.2. Compensatory Mechanisms Impairement

Heart failure is characterized by a decreased cardiac output, which leads to a decreased mean arterial pressure (MAP) and decreased tissue perfusion. The body thus reacts by using several compensatory mechanisms to bring the MAP back to normal in an attempt to maintain adequate tissue perfusion. These actions include the Frank–Starling mechanism, neurohormonal activation, and ventricular remodeling. However, while initially beneficial, in the long-term, these adaptative mechanisms become maladaptive when trying to sustain adequate cardiac performance and, thus, worsen heart failure in a vicious cycle.

A decrease in cardiac output leads to the stimulation of the neuroendocrine system which releases epinephrine, norepinephrine, endothelin 1, and vasopressin. These molecules have direct effects on the heart (increased heart rate and contractility) and on the peripheral vasculature (vasoconstriction), which increases stroke volume and total peripheral resistance and, thus, cardiac output and the MAP [3,5].

Decreases in CO also stimulate the renin angiotensin aldosterone system by promoting sodium and water retention, which stimulates the release of vasopressin and increases vasoconstriction. The RAAS also releases angiotensin II, which has been shown to increase myocardial cellular hypertrophy and cellular interstitial fibrosis [5,19]. In the long-term, these effects result in ventricular remodeling, which further deteriorate myocardial dysfunction. However, most therapies aiming to attenuate HF conditions counteract the deleterious effects of these compensatory mechanisms [20].

### 2.3. Regulatory Mechanisms

By consistently filtering blood from glomeruli into renal tubules, the kidneys have a crucial role in maintaining the balance of body fluids. Through reabsorption and secretion, water and solutes are exchanged between the tubules and peritubular capillaries, thus leading to the excretion of excess water, solutes, and waste products in the urine. However, these processes are regulated by multiple intrinsic and neurohormonal feedback mechanisms. In what follows, we will describe the key physiological mechanisms for these control mechanisms regulating the cardiac and renal function.

#### 2.3.1. Vasopressin

Vasopressin, also known as antidiuretic hormone (ADH) or arginine vasopressin (AVP), is a hormone that is secreted by the posterior pituitary gland due to changes in osmolarity and volume status. In the body, vasopressin acts on the kidney and blood vessels. It helps prevent the loss of water from the body by reducing urine output and serving the kidney in reabsorbing water in the late distal tubule and collecting duct [21,22]. Vasopressin also induces vasoconstriction. Together, these two mechanisms serve to increase effective arterial blood volume and increase blood pressure to maintain tissue perfusion.

#### 2.3.2. Pressure Natriuresis

The principle of pressure natriuresis is that increases to renal perfusion pressure result in reductions in the reabsorption of sodium in the tubules and increases to the amount of sodium excreted by the body [23]. In response to increases in renal arterial pressure, the kidneys react to maintain normal levels of sodium and systemic arterial pressure by increasing the excretion of sodium and reducing the amount of extracellular fluid. Elevated renal perfusion pressure causes an increase in renal interstitial hydrostatic pressure (RIHP), thus ultimately impacting tubular reabsorption through several ways, including changes to the permeability of tight junctions to sodium in the proximal tubules, the redistribution of apical sodium transporters, and the release of autacoids [23,24]. The attenuation of pressure natriuresis can be achieved by preventing an increase in RIHP in response to a rise in renal perfusion. By directly acting on the transport of sodium in renal tubules, renal autacoids, including nitric oxide, prostaglandin E2, kinins, and angiotensin II, may regulate pressure natriuresis.

#### 2.3.3. Renin Angiotensin Aldosterone System

The renin angiotensin aldosterone system (RAAS) plays an important role in controlling several key factors in the cardio–renal system. These include blood volume, sodium reabsorption, potassium secretion, water reabsorption, and vascular tone [25]. Thus, RAAS impacts acutely on blood pressure. Other functions of the RAAS include inflammation, apoptosis, and fibrosis [26]. The RAAS consists of three essential components (renin, angiotensin II, and aldosterone) that act in concert to increase arterial pressure in response to reduced renal blood pressure, decreased salt delivery to the distal convoluted tubule, or beta-agonism [25].

The cascade starts from the production of renin by the activation of the juxtaglomerular cells in the afferent arterioles of the kidney, thereby cleaving the inactive prorenin to renin. These cells are activated when there is a reduction in blood pressure, beta-activation, or when macula densa cells detect a decreased sodium load in the distal convoluted tubule [27,28]. Upon release into the bloodstream, renin targets angiotensinogen in the plasma. This cleavage of angiotensinogen by renin forms angiotensin I (physiologically inactive), which is further converted to angiotensin II by an enzyme called angiotensin-converting enzyme (ACE). ACE is located predominantly in the vascular endothelium of the kidneys and lungs [25]. Angiotensin II binds to angiotensin II type I (AT1) receptors, thus causing various effects on the kidneys, adrenal cortex, arterioles, and brain [29]. When bound to AT1 receptors, angiotensin II stimulates renal and systemic vasoconstriction, raises sodium reabsorption by increasing the activity of NHE3 in the proximal tubule of the kidney, and promotes inflammation and fibrosis [25,30]. When bound to AT2 receptors, it induces vasodilation and natriuresis.

Angiotensin II also stimulates the release of aldosterone from the zona glomerulosa of the adrenal cortex [31]. Aldosterone, in turn, promotes sodium reabsorption and potassium excretion at the distal tubule and collecting duct of the nephron. Angiotensin II can also be metabolized by aminopeptidase A (APA) to form angiotensin III, which binds to AT1 receptors to induce its effects. The further cleavage of angiotensin III by alanyl aminopeptidase N (APN) generates angiotensin IV, which produces cardioprotective effects through binding to AT4 receptors. These effects include increases in natriuresis and NO production and reductions in vasoconstriction and inflammation [32].

Recent studies have documented the concept of a counter regulatory RAAS with opposing effects on cardiovascular function [32]. This counter regulatory system involves angiotensin-converting enzyme 2 (ACE2) and neprilysin (NEP). By cleaving angiotensin II, these enzymes produce angiotensin 1-7 (Ang 1-7). Angiotensin I can also be cleaved by ACE2 and NEP to produce angiotensin 1-9 (Ang 1-9) and Ang 1-7, respectively. Ang 1-7 bind to the proto-oncogene MAS receptors, which lead to vasodilation, antihypertensive, and antifibrosis effects, while Ang 1-9 can activate AT2 receptors to trigger natriuresis and NO production, thus mediating vasodilatory effects and reducing blood pressure [26]. Furthermore, angiotensin II acts on the brain where it binds to the hypothalamus, thus stimulating thirst and increasing water intake [25]. In the end, the net effects of all these interactions are an increase in total body sodium, total body water, and vascular tone.

#### 2.3.4. Myogenic Response and Tubuloglomerular Feedback

To maintain high glomerular filtration rate (GFR) in the face of blood pressure fluctuation, the kidney must maintain constant renal blood flow and GFR. This is achieved through renal autoregulation, which is a process mediated by the combined and interacting contributions of two mechanisms: a faster myogenic response and a slower tubuloglomerular feedback (TGF) [33]. Specific to the kidneys, TGF is a regulatory mechanism that responds to an increase in the luminal concentration of NaCl at the macula densa in the early distal tubule and causes vasoconstriction of the afferent arteriole [33].

The smooth muscles that form the afferent arteriole walls constrict in response to elevated pressure and dilate in response to decreased pressure; this is known as the myogenic response and is present in almost all terminal vessels in the body [34]. However, the myogenic response of the renal afferent arteriole is distinct from other vascular beds due to its capability to buffer large pressure disturbances and its short response.

#### 2.3.5. Renal Sympathetic Nerve Activity

The kidneys are densely innervated with renal afferent and efferent nerves to communicate with the central nervous system. Renal sympathetic nerve activity (RSNA) has an important role in renal disease-associated hypertension and in the modulation of fluid homeostasis [35]. RSNA increases renal vasculature resistance, as well as proximal tubule sodium reabsorption, and stimulates renin release from the juxtaglomerular cells. It has been shown that an increase in arterial pressure and right atrial pressure causes the RSNA to decrease [36,37]. RSNA is commonly increased in pathophysiological conditions such as hypertension, heart failure, and chronic and end-stage renal function [38]. Increased RSNA raises blood pressure and can contribute to the deterioration of renal function. Various clinical trials of catheter-based renal sympathetic denervation in the management of resistant hypertension have found promising results associated with significant falls in blood pressure and renal protection [39], thus highlighting the potent role of renal efferent and afferent nerves and their utility as a therapeutic target.

#### 2.3.6. Natriuretic Peptides

Natriuretic peptides are a family of three structurally related hormones/paracrine factors, including atrial natriuretic peptide (ANP), brain or B-type natriuretic peptide (BNP), and C-type natriuretic peptide (CNP) [40]. ANP and BNP, which are predominantly produced by cardiomyocytes, are found in the atria and ventricles, respectively, and are released following an atrial or ventricular stretch [3]. CNP is found more in the central nervous system and peripheral tissues [41]. These hormones demonstrate pleiotropic cardiovascular and metabolic properties including vasodilation, natriuresis, and reduction of the RAAS [42]. They exert their effects by increasing the amounts of cyclic guanosine monophosphate (cGMP) that circulate in target tissues [43]. BNP has become clinically useful in the diagnosis of heart failure; its elevated level is thought to be one of the first signs of HF and is used to follow the progression of the disease [41].

## 3. Treating Heart Failure

Treatments for heart failure include improvements of lifestyle in addition to pharmacological therapies. Patients are encouraged to lose excess weight, abstain from tobacco and alcohol use, and perform physical exercises as tolerated. Pharmacological and medical approaches to treating HF include targeting the risk factors of heart failure, such as hypertension, diabetes, dyslipidemia and arrhythmias through pharmacological interventions, as well as surgical management, including cardiac resynchronization therapy, coronary revascularization, surgical ventricular remodeling implantation, and heart transplantation [44,45]. In this review, we focus solely on pharmacological management and the use of mathematical modelling to improve treatment approaches.

### 3.1. Treatment History

Early treatments of heart failure consisted of a combination of digitalis and diuretics, which relieved acute symptoms but failed to alter the long-term survival of patients once they were released from the hospital [46]. Digitalis (e.g., digoxin, digitoxin, medigoxin, etc.) acts to enhance inotropy of the cardiac muscle and also reduces the activation of the sympathetic nervous system and RAAS [47,48]. Counterintuitively, digitalis toxicity showed a risk of sudden cardiac death by inducing fatal cardiac arrhythmias [47]. It is commonly believed that digitalis, by increasing cardiac intracellular calcium (${\mathrm{Ca}}^{2+}$), provides both inotropic and arrhythmogenic effects [47]. As the compensatory mechanisms of HF involve fluid retention, diuretics were included to relieve pulmonary congestion and peripheral edema [12] by reducing extracellular fluid volume. They act by reducing sodium reabsorption at different sites in the nephron, thus resulting in an increase in urinary sodium and water losses. They further showed an improvement in exercise tolerance [49], which is critical for patients’ overall well-being.

Currently, digitalis is rarely used in the management of HF [50]. Studies have shown that it fails to reduce the overall mortality [51], probably due to its toxicity that provokes additional deaths. Thus, the failure of digitalis-based approaches to treat HF coincided with the introduction of antihypertensive therapies that counteract the deleterious effects of the neurohormonal mechanisms, particularly the RAAS, as was discussed previously. These drugs include angiotensin-converting enzyme (ACE) inhibitors [9,10] that block the conversion of angiotensin I to angiotensin II, thereby reducing RAAS activation. Similarly, mineralocorticoid receptor antagonists (MRA), which are aldosterone antagonists, directly inhibit RAAS to offset its effects. Angiotensin receptor blockers (ARB), which directly block the angiotensin II receptors, are also used in patients who cannot tolerate ACE inhibitor therapies [11], in addition to beta-blockers that reduce the heart’s workload, thus allowing for more efficient contractions and also producing vasodilatory effects [52,53]. For example, by reducing the $\beta_{1}$ receptors, beta-blockers alter the effects of catecholamines, epinephrine, and norepinephrine, thereby reducing the heart rate with less contraction, which, thus, contributes to lower blood pressure and prevents arrhythmias [54].

### 3.2. Next Generation of Heart Failure Treatments: SGLT2 Inhibitors

Until recently, the pharmacological mainstay of established HF-rEF therapy has been a three-drug approach with renin angiotensin system inhibitors, beta-blockers, and mineralocorticoid antagonists [55]. However, a new class of drugs (i.e., SGLT2 inhibitors) has emerged as possible fourth drug in front-line therapy with promising benefits. SGLT2 inhibitors were originally developed to treat type 2 diabetes mellitus [56,57,58]. However, unlike other antihyperglycemic agents, they have been shown to reduce the risk of hospitalization due to HF-rEF and serious renal outcomes in patients with diabetes [59,60,61]. These benefits suggested that SGLT2 inhibitors could add cardioprotective and renoprotective effects in patients with established heart failure, including those with and without diabetes [62,63]. However, the mechanisms with which these inhibitors induce these cardiorenal benefits are poorly understood. Below, we describe some of the potential pleiotropic mechanisms of action.

Improvement of glucosuria and natriuresis: The primary mechanism of SGLT2 inhibitors is the inhibition of the effects of SGLT2 in the proximal tubule of the kidney, thus leading to glucosuria and natriuresis (i.e., an increase in urinary excretion of glucose and sodium, respectively) [64,65,66]. Glucosuria results in improved glycemic control, weight loss, and a subsequent osmotic diuresis. Natriuresis leads to a decrease in blood and plasma volume, as well as to hemoconcentration, and further contributes to a decrease in both preload and afterload, and in mean arterial pressure [65].

Inhibition of sodium hydrogen exchanger (NHE): Another benefit of SGLT2 inhibitors is the inhibition of the NHE transporters in the kidney (NHE3) and in the myocardium (NHE1). In the kidney, the inhibition of NHE in the proximal tubule may increase the delivery of sodium to the distal tubule and help preserve renal perfusion [67]. In the myocardium, it can lower myocardial ${\mathrm{Na}}^{+}$ levels which, in turn, decrease the cytosolic calcium to prevent cell death. The subsequent increase in mitochondrial ${\mathrm{Ca}}^{2+}$ is also beneficial for the prevention of heart failure [68].

Decrease in oxidative stress: An important pleiotropic effect of SGLT2 inhibitors includes the decrease in reactive oxygen species (ROS), either through direct action (i.e., reducing the interaction between SGLT1 and NADPH oxidase activity via a mixed SGLT2 and SGLT1 inhibitory action), or through an indirect action (i.e., improved glycemic control) [66]. Excessive cardiac mitochondrial ROS has been shown to be a strong contributor to contractile dysfunction in both animal models of heart failure and in humans [64,69].

Vascular function improvement: SGLT2 inhibition has been shown to improve vascular function by reducing endothelial cell activation and dysfunction and inducing direct vasodilatation. Together, they decrease vascular resistance [64,70,71,72].

Reduction in sympathetic nervous system (SNS) overactivity: It is believed that SGLT2 inhibitors may also reduce central sympathetic overactivity through the suppression of renal afferent signaling to the brain [64,73]. Sustained sympathetic overactivity can result in increased arterial stiffness, endothelial dysfunction, and altered renal sodium and water balance. Together, these pathophysiological mechanisms ultimately lead to fluid retention and edema [73]. The renal renin angiotensin system, or RAS, is also a target of SGLT2 inhibition, thus causing increases to circulating natriuretic peptide levels through inhibited renal neprilysin activity. These coinciding mechanisms ultimately reduce sympathetic nervous system activity, thus balancing the natriuretic peptide/soluble guanylate cyclase (sGC)/cyclic guanosine monophosphate (cGMP) pathway and the RAS pathway, which provide cardiovascular and renal protection [73].

Having discussed the pathophysiology and treatment strategies for HF, we will briefly describe in the following sections key aspects of existing quantitative systems pharmacology models of the cardiorenal system that serve to (1) quantify clinically relevant outcomes in response to therapies, and (2) provide a predictive framework to improve therapeutic scheduling and strategies, including preclinical drug development.

## 4. Mathematical Modelling of the Cardiorenal Function

Mathematical and computational modeling have played an increasingly important role in advances in medicine. Such models provide a framework that can guide the prediction of biological processes and define the complex landscape of disease, thus eventually leading to more effective and reliable methods for disease diagnosis, risk stratification, and therapy [74]. Because the renal and cardiac systems operate largely on flux–balance (as described above), several mathematical modelling strategies have been developed to predict the pathophysiology of heart failure and other related diseases, such as hypertension. These models overall use ordinary differential equations and algebraic equations.

### 4.1. Modeling Renal Function

The kidney is an important organ that is responsible for filtering blood into urine to remove waste products (e.g., urea, creatinine, amino acids) from the bloodstream. It also regulates water balance, electrolyte balance (i.e., sodium, glucose, chloride, potassium, magnesium, phosphate, etc.) and acid–base balance through the mechanisms of filtration, reabsorption, secretion, and excretion [75,76]. These regulatory functions of water and ${\mathrm{Na}}^{+}$ balance, together with its ability to secrete hormones and vasoactive factors or substances (e.g., renin), contribute to the control of blood pressure and volume status [75].

Structurally, the kidney can be modeled in terms of its functional units [77,78], i.e., its approximately 1 million nephrons [79], or, more simplistically, we can consider the kidneys as one single large nephron [80,81,82]. Modeling multiple nephrons is advantageous, as diseases such as glomerulosclerosis, which is accompanied with a nephron loss [77], can affect a number of nephrons. Each nephron constitutes a glomerulus (i.e., afferent arteriole, glomerular capillaries, and efferent arteriole) and tubule (i.e., proximal tubule (PT), loop of Henle (LoH), distal convoluted tubule (DCT), and connecting tubule and collecting duct (CNT/CD)) [34,83].

Blood enters the kidney through the renal artery, which first divides into segmental arteries, followed by further branching (interlobar, arcuate, and interlobular arteries) [34,84] to finally reach the afferent arteriole. In [79], all of the arteries prior to the afferent arteriole are lumped together to form the pre-afferent. Blood then passes through the glomerular capillaries (located in the Bowman capsule) where it is filtered before continuing its way through the efferent arteriole and peritubular capillaries, and it then finally moves out of the kidney via the renal venous network.

The first step in urine formation is the filtration of blood plasma across the glomerular capillary walls. Once blood is filtered, sodium, chloride, glucose, water, and other electrolytes follow the tubules (i.e., PT, LoH, DCT, CNT/CD) before being excreted as urine. Along this course, some substances are reabsorbed back to the circulation, whereas others (e.g., hydrogen, potassium, urea, creatinine, toxins) are secreted from peritubular capillaries into the tubular lumen [85,86,87]. The urine that is then formed derives from these three processes of filtration, reabsorption, and secretion. Existing models [88,89,90] try to capture the flow rates of solutes and water along each segment of the tubule by taking into account their reabsorption rates to finally predict their composition into the urine (see Figure 1).

Over the years, several attempts have been made to revise the first extensive model published by Guyton et al. in 1972 [80] (e.g., [79,81,82,88,91]). Briefly, these models aim to capture the key physiological processes involved in renal function and in maintaining sodium and water homeostasis. Hallow et al. [89] and Layton et al. [88] extended these previous models by considering a more representative kidney physiology, which included more solute transporters (e.g., SGLT1, SGLT2, NHE3) in the nephron tubule to account for more solute transport along the tubule [88], such as glucose. As endpoints, these models compute the flow rates of water and solutes along each tubular segment by considering their reabsorption rates to finally get their urine composition.

As a representative example, here, we present a portion of the Hallow et al. model [89] describing the filtration, reabsorption, and excretion of glucose and sodium along the tubule as illustrated in Figure 1.

#### 4.1.1. Glucose Transport along the Nephron

The renal PT consists of two segments: the proximal convoluted tubule (S1 and S2 segments) and the proximal straight tubule (S3 segment) [92,93]. Under normal conditions, around 90*–*97% of glucose is reabsorbed by the SGLT2 located in the S1 and S2 segments of the PT, while the remaining 3*–*10% is reabsorbed by the SGLT1 in the S3 segment [93,94,95]. This implies that when blood glucose concentration is up to 9 mmol/L [96], all glucose is reabsorbed and there is no urinary glucose excretion (UGE). First, glucose is filtered freely via the glomerulus, thus resulting in a single nephron filtered glucose load of:(1)$$\begin{array}{c} \Phi_{glu,filtered}=SNGFR*C_{glu}, \end{array}$$
where $C_{glu}$ is the plasma glucose concentration.

Hallow et al. [89] described the glucose load reabsorption in S1–S2 and in S3 as:(2)$$\begin{array}{c} \Phi_{glu,reabs,S12}=\min\left(\Phi_{glu,filtered},\;R_{glu,S12}*L_{pt,S12}\right), \end{array}$$
(3)$$\begin{array}{c} \Phi_{glu,reabs,S3}=\min\left(R_{glu,S3}*L_{pt,S3},\;\Phi_{glu,filtered}-\Phi_{glu,reabs,S12}\right), \end{array}$$
where $R_{glu,S12}$ and $R_{glu,S3}$ are the rates of glucose reabsorption per unit length of the combined S1–S2 and S3 segments, respectively. $L_{pt,S12}$ and $L_{pt,S3}$ are the length of PT per S1–S2 and S3 segments, respectively. However, glucose is only reabsorbed in the PT segment, so any unabsorbed glucose flows through the rest of the tubule and is excreted at a rate that is defined as follows:(4)$$\begin{array}{c} R_{UGE}=\Phi_{glu,out-PT}=\Phi_{glu,filtered}-\Phi_{glu,reabs,S12}-\Phi_{glu,reabs,S3}.\; \end{array}$$

#### 4.1.2. Sodium Transport along the Nephron

Sodium is also freely filtered in the glomerulus, and the single nephron filtered ${\mathrm{Na}}^{+}$ load is:(5)$$\begin{array}{c} \Phi_{glu,filtered}=SNGFR*C_{glu}, \end{array}$$
where $C_{Na}$ is the plasma ${\mathrm{Na}}^{+}$ concentration.

Under normal circumstances, the PT is responsible for reabsorbing 65*–*70% of filtered sodium and water [92,97]. This sodium reabsorption process involves many transporters (e.g., SGLT2, SGLT1, NHE3) whose expressions vary across different segments of the PT.

Since the reabsorption of ${\mathrm{Na}}^{+}$ and glucose through the SGLT2 and SGLT1 occurs at a 1-to-1 molar ratio and at a 2-to-1 molar ratio, respectively, the rates of ${\mathrm{Na}}^{+}$ reabsorption in these transporters can be obtained from the rates of glucose reabsorption [89,98]:(6)$$\begin{array}{c} \Phi_{Na,reabs-SGLT2}=\Phi_{glu,reabs,S12},\; \end{array}$$
(7)$$\begin{array}{c} \Phi_{Na,reabs-SGLT1}=2\Phi_{glu,reabs,S3}. \end{array}$$

Thus, the total $Na^{+}$ PT reabsorption is given by:(8)$$\begin{array}{c} \Phi_{Na,reabs-PT}=\Phi_{Na,reabs-SGLT2}+\Phi_{Na,reabs-SGLT1}+\Phi_{Na,filtered}\left(\eta_{Na,reabs-PT,NHE}+\eta_{Na,reabs-PT,other}\right), \end{array}$$
where $\eta_{Na,reabs-PT,NHE}$ and $\eta_{Na,reabs-PT,other}$ are the fractional rates of PT ${\mathrm{Na}}^{+}$ reabsorption through the NHE3 and through mechanisms other than the SGLT2 and SGLT1, respectively.

Other transporters of ${\mathrm{Na}}^{+}$ may include the sodium bicarbonate cotransporter (NBC), the sodium phosphate cotransporter 2 (NaPi2), sodium amino acid transporters, sodium-potassium-adenosine triphosphatase ($Na^{+}/K^{+}\mathrm{ATPase}$) [92,97].

Hence, $Na^{+}$ flow out of the PT is given by:(9)$$\begin{array}{c} \Phi_{Na,out-PT}=\Phi_{Na,filtered}-\Phi_{Na,reabs-PT}.\; \end{array}$$

Besides these renal processes of reabsorption and excretion, sodium and water balance are also regulated by osmotic exchange between blood and interstitial fluid, as well as non-osmotic ${\mathrm{Na}}^{+}$ storage in peripheral tissues such as skin, muscle, and brain [99,100,101]. In the next section, we will discuss the cardiovascular hemodynamics and see how they are linked to the renal function.

### 4.2. Modeling Cardiovascular Function

Heart failure is characterized by an excess fluid accumulation in the interstitium, which leads to peripheral and pulmonary edema. Treatment therefore aims to relieve this congestion. Hallow et al. [89] extended previous studies focused on systemic water and sodium dynamics [80,91] by considering the role of non-osmotic ${\mathrm{Na}}^{+}$ storage in the peripheral tissues such as skin, muscles, and brain [99,100,101]. They modeled ${\mathrm{Na}}^{+}$ and water exchanges between blood, interstitial fluid, and peripheral tissues using a three-compartment model [102]. Their model allowed for the evaluation of the impact of water clearance and osmotically inactive ${\mathrm{Na}}^{+}$ storage on blood and interstitial volume, thereby enabling the prediction of volume responses (which are critically beneficial in heart failure) to pharmacological treatments [102].

In the work of Hallow et al. [89], cardiac and renal function were coupled through blood volume, which is regulated by the kidney through the control of sodium and water excretion, and by the MAP, which is calculated from [80,91] and is a key determinant of renal perfusion and glomerular filtration rates in the kidney model (see Figure 2).

In this portion of the Hallow et al. model [89], exchanges between blood and interstitial fluid occur along a ${\mathrm{Na}}^{+}$ concentration gradient. This implies that sodium moves from a more concentrated environment to a less concentrated one, whereas water moves in the opposite direction, from a less concentrated (i.e., more diluted) to more concentrated (i.e., less diluted) environment. Their model does not consider exchange with the intracellular space. It is assumed that ${\mathrm{Na}}^{+}$ concentrations in the blood and interstitial fluid quickly equilibrate. They also assumed that when interstitial ${\mathrm{Na}}^{+}$ concentrations exceed equilibrium levels, ${\mathrm{Na}}^{+}$ moves out of the interstitium to be stored in the peripheral ${\mathrm{Na}}^{+}$ compartment, where it is not osmotically active. However, it is not possible to store ${\mathrm{Na}}^{+}$ indefinitely, so there exists a limit $Na_{stored,max}$ on the amount of stored ${\mathrm{Na}}^{+}$. Blood volume ($V_{b}$) and interstitial fluid volume ($V_{IF}$) are modeled in a comparable manner, where the difference in ${\mathrm{Na}}^{+}$ concentration between blood and interstitium drives the transfer of water between these two compartments. These dynamics are described by the following equations:(10)$$\begin{array}{c} \Phi_{Na,stored}=Q_{Na,stored}*\frac{Na_{stored,max}-Na_{stored}}{Na_{stored,max}}\left({\left[Na\right]}_{IF}-{\left[Na\right]}_{IF,ref}\right)\; \end{array}$$
(11)$$\begin{array}{c} \frac{d}{dt}\left(Na_{blood}\right)=\Phi_{Na,intake}-\Phi_{Na,excretion}-Q_{Na}\left({\left[Na\right]}_{blood}-{\left[Na\right]}_{IF}\right)\; \end{array}$$
(12)$$\begin{array}{c} \frac{d}{dt}\left(Na_{IF}\right)=Q_{Na}\left({\left[Na\right]}_{blood}-{\left[Na\right]}_{IF}\right)-\Phi_{Na,stored}\; \end{array}$$
(13)$$\begin{array}{c} \frac{d}{dt}\left(Na_{stored}\right)=\Phi_{Na,stored}\; \end{array}$$
(14)$$\begin{array}{c} \frac{d}{dt}\left(V_{blood}\right)=\Phi_{water,intake}-\Phi_{water,excretion}-Q_{water}\left({\left[Na\right]}_{IF}-{\left[Na\right]}_{blood}\right)\; \end{array}$$
(15)$$\begin{array}{c} \frac{d}{dt}\left(V_{IF}\right)=Q_{water}\left({\left[Na\right]}_{IF}-{\left[Na\right]}_{blood}\right)\; \end{array}$$
(16)$$\begin{array}{c} {\left[Na\right]}_{blood}=\frac{Na_{blood}}{V_{blood}}\; \end{array}$$
(17)$$\begin{array}{c} {\left[Na\right]}_{IF}=\frac{Na_{IF}}{V_{IF}}. \end{array}$$

As previously noted, heart failure is also characterized by ventricular remodeling in response to changes in pressure and volume overload. More sophisticated cardiac models [103,104,105] integrate this myocardial adaptive response, which leads to an increase in the myocyte length and/or diameter in addition to the regulatory mechanisms discussed above, which are essential for maintaining water and solute homeostasis. These remodeling patterns impact cardiac function, which in turn alters renal function through changes in renal perfusion, thus leading to changes in blood and/or interstitial fluid volume.

**Figure 1 pharmaceutics-15-01002-f001:**
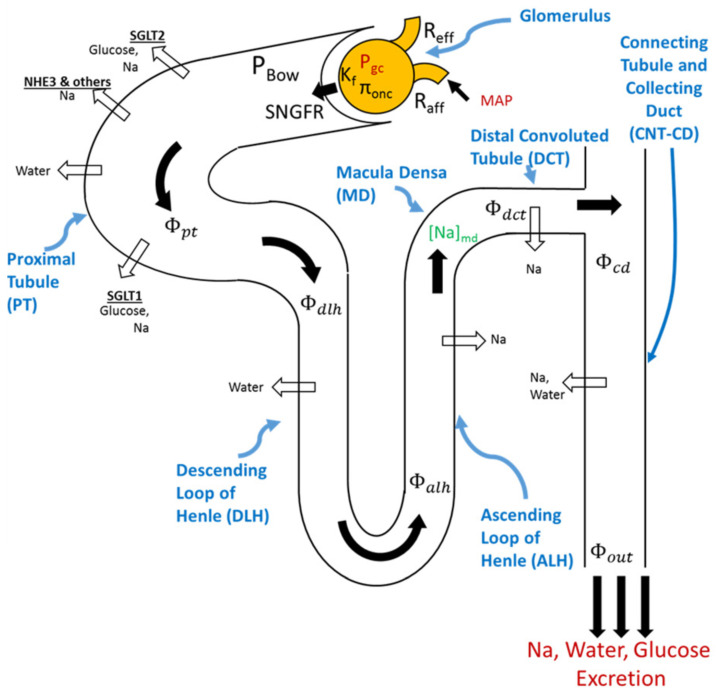
**Schematic overview of the filtration, reabsorption, and flow rates in the nephron.** Glomerular filtration is modeled according to Starling’s law. ${\mathrm{Na}}^{+}$ and water are reabsorbed at different fractional rates in the proximal tubule, loop of Henle, distal convoluted tubule, and connecting tubule/collecting duct. Glucose and$\;{\mathrm{Na}}^{+}$ reabsorption are coupled through sodium–glucose cotransporter (SGLT)2 and SGLT1 in the proximal tubule. Adapted from Hallow et al. [105] under Creative Commons CC BY.

**Figure 2 pharmaceutics-15-01002-f002:**
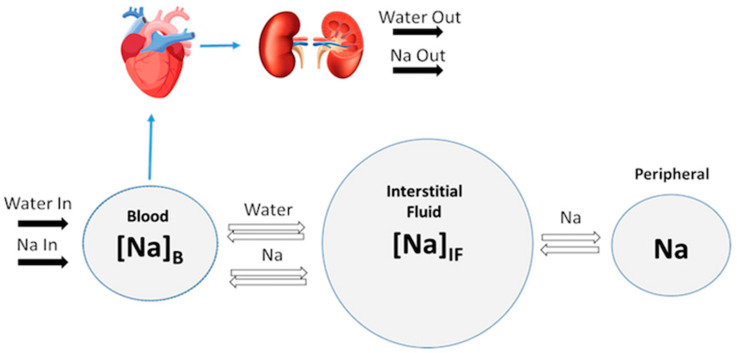
**Schematic representation of the model’s different compartments and their link to the renal function.** The balance between ${\mathrm{Na}}^{+}$ and water excretion and intake determine blood volume and ${\mathrm{Na}}^{+}$ concentration. ${\mathrm{Na}}^{+}$ and water move between the blood and interstitial fluid across a concentration gradient, and ${\mathrm{Na}}^{+}$ may be sequestered non-osmotically in a peripheral storage compartment. Blood volume determines venous return and cardiac output, which, together with total peripheral resistance, determine mean arterial pressure and subsequent renal perfusion pressure, thus closing the loop. Adapted from Hallow et al. [105] under Creative Commons CC BY.

## 5. Modeling and Optimizing Treatments

As mentioned previously, one can use QSP models to predict how pharmacological interventions affect the dynamics of biomarkers and clinical endpoints. These models mechanistically describe pathophysiological processes and model dynamic quantitative physiological endpoints [16,106]. Once a model structure is established, the next step in the process is to calibrate and validate it by comparing predictions with preclinical and clinical data. This is to ensure that the model can capture and explain observed clinical trial biomarkers and response variability, which are critical to address prior to applying the model to clinical development questions.

In both academia and the industry, QSP models provide the backbone of virtual clinical trials [107,108,109], with the goal of predicting the effectiveness of drugs on individuals with specific medical conditions (e.g., diabetic, or non-diabetic HF patients). They do this through the generation of virtual patient cohorts [110] that can be leveraged to understand pathophysiological mechanisms and heterogeneous treatment responses. To create a virtual patient cohort that accurately represents the clinical population, a set of model parameters are chosen to be varied, while the remaining parameters are maintained at their default values [111]. The set of selected parameters aims to capture the interindividual variations observed in real patients.

After validating the QSP platform by conducting virtual clinical trials, virtual patients are also used to predict the additive effects of the drugs and to explore dose optimization. Considering more than one therapy dramatically increases the complexity of predicting the optimal drug order and schedule for individual patients, and competing evidence supports simultaneous, sequential, or alternating treatment plans [112]. Designing optimal therapeutic schedules that benefit the individual will likely require personalized medicine treatment strategies that utilize both mathematical and clinical triage to assess what is both theoretically optimal and most practical [112].

For example, CKD patients represent a critical clinical subgroup that need to be treated with care regarding their drug regimen dosage due to a decreased renal function. In fact, many drugs are eliminated by the kidney and therefore require dose adjustment in patients with renal insufficiency [113]. In these types of patients, the doses of many drugs must be reduced to avoid toxicity. Many aspects of drug disposition, such as bioavailability, absorption, protein binding, distribution volume, metabolism, and clearance, are altered in CKD [114]. CKD also affects multiple organ systems and can alter the dose–exposure response relationship of certain classes of compounds, particularly in advanced stages of CKD (i.e., stages IV and V) [114,115,116,117]. Thus, drug developers need to understand how CKD affects the PK and PD of drugs, particularly when it is likely to change the PK of a drug or its active/toxic metabolite. The United States Food and Drug Administration (FDA) recommends following such guidance [118].

However, despite the rise of QSP modeling, there are still conceptual and organizational barriers hindering the strategic use of this type of computational modeling as described in [119]. For example, a QSP model may be seen to be not immediately comprehensible to non-mathematicians, which may lead to an overall hesitancy from decision makers in the pharmaceutical industry when approaching QSP modeling [119]. Another common objection arises with respect to the sense of incompleteness of QSP models, as, in the case of experimental or clinical models, they cannot comprise all the factors influencing any given system. In fact, all models are (and must be) simplifications of reality built from assumptions, facts, and mental models that are already being used in decision making [119]. Thus, a more relevant question is not whether the models are 100% representative and complete, but rather how wrong do they have to be to not be useful [120]. Indeed, even with limited detailed information and low-resolution models, QSP modeling can reveal (and propose for further exploration) unexpected and emergent systemic dangers and opportunities that may not be otherwise ascertainable from wet lab experiments or clinical trials.

## 6. Considering Sex-Specific Differences in Heart Failure

An important aspect to consider in all treatment scenarios is sex differences between males and females and their impact on health and disease; this is crucial for the development of effective therapies for broad populations. Encouragingly, mathematical modeling and quantitative systems pharmacology are particularly well-suited to contribute to improving sex-based treatment administration. Sexual dimorphism has been observed in numerous physiological systems, including the brain, the activities of the stress and immune systems, and metabolic and cardiovascular functions [121,122,123]. Here, we will review the differences seen in cardiovascular and kidney function [124,125] that result in, e.g., differences in the regulation of blood pressure between male and females [126], which ultimately affect the responses seen in antihypertensive therapies between males and females [127] through the lens of QSP models.

Until recently, most mathematical models that extended Guyton’s work [80] have been sex neutral [79,81,128]. However, male and female rodents exhibit notable differences in the expression pattern of membrane transporters in the kidney [129,130]. Therefore, Layton et al. developed computational epithelial transport models of rats [131,132,133] and humans [134]. Together, these models predicted that females have a lower activity of NHE3, a major sodium transporter along the proximal tubule, which results in a shift in the sodium transport load to the distal tubule [131]. The distal tubular segments are responsible for finely adjusting sodium and water excretion, and that shift in sodium transport in females may enable them to handle acute salt loads more efficiently [131,135]. Furthermore, this phenomenon may also contribute to better blood pressure control in females.

Studies have demonstrated that female hypertensive rats have a greater availability of NO [136,137]. NO is known to regulate vascular tone and hemodynamics, as well as inhibit the reabsorption of sodium in the tubules, which ultimately promote diuresis and natriuresis and, thus, contribute to blood pressure control. Females exhibit higher levels of NO bioavailability due to greater NO production compared to males [138], as is evidenced by the increased expression of NO synthesis (NOS) 1 and NO3 in the renal vasculature of females, as well as by their lower levels of NO scavenging by superoxide [135].

These physiological differences may provide a better understanding of both sexes’ abilities to compensate for blood pressure dysregulation, which arises from various pathophysiological mechanisms, such as increased systemic vascular resistance, increased afferent arteriole resistance, or overactivity of the RAS or renal sympathetic nerve activity (RSNA). Leete and Layton’s work [126] suggests that the severity of hypertension induced may vary significantly between the sexes for a given pathophysiological perturbation. Their simulations further indicate that the stronger RSNA-mediated regulation of afferent arteriole tone in females is primarily responsible for their resistance to developing hypertension.

Given these findings, Layton et al. extended their model to provide a biological rational for observed differences in male and female drug efficacy between ARB and ACE inhibitors (both of which target RAS) for the treatment of hypertension [127]. Their predictions allowed them to conclude that these differences can be attributed to a potential vasodilatory, and, thus, potentially blood pressure lowering, effect of the binding angiotensin II to the receptors AT2, which was assumed to be present only in females. This may explain the higher efficacy of ARB compared to ACE inhibitors because, in addition to the primary action of lowering blood pressure, the administration of ARB inhibitors, but not ACE inhibitors, increases the level of angiotensin II that binds into the AT2 receptors, inducing vasodilation [127,135]. Since we know that chronic hypertension is a leading risk factor for heart failure, and that all pathophysiological perturbations found in hypertension are also altered in the case of HF, these studies point the way for a better management of HF by considering sex-based therapies between males and females.

## 7. Discussion

HF is a globally prevalent medical condition that arises due to structural defects in the heart that hinder its ability to pump blood efficiently at normal cardiac pressures, thereby failing to meet the body’s metabolic demands. These irregularities are frequently caused by ischemic heart disease, hypertension, and diabetes. Therapeutic approaches involve the use of diuretics to remove excess fluids, the inhibition of hyperactive neurohormonal systems (e.g., antihypertensive medications), and the improvement of cardiac contractility (e.g., digitalis, beta-blockers). Currently, there is no evidence-based pharmacological therapy for HF-pEF, possibly due to the type of hypertrophy we have (i.e., concentric hypertrophy), which results in an abnormal filling pattern rather than a volume overload, as is observed in HF-rEF.

SGLT2 inhibitors, which are a novel class of antihyperglycemic drug, have been surprisingly found to significantly reduce hospitalizations for HF and cardiovascular mortality in patients with HF-rEF, which is potentially due to their numerous pleiotropic effects. Therefore, further studies have been conducted to examine the efficacy of these inhibitors in patients with HF-pEF (EMPEROR-Preserved trial [139]). The outcomes of this trial showed encouraging results, as empagliflozin reduced the combined risk of hospitalization for HF or cardiovascular death in patients with HF-pEF, regardless of whether they had diabetes. These findings imply that SGLT2 inhibitors may be the first class of medication to enhance cardiovascular outcomes in individuals with HF-pEF. Given the recent rise in the use of SGLT2 inhibitors, there is a need for an improved understanding of their effects on the entire body and for tailoring their use in particular patient populations. In this case, mathematical and QSP modelling is well-positioned to respond.

Beyond specific treatments, there have also been documented instances of sexual dimorphism in cardiovascular and renal function, thereby leading to disparities in blood pressure regulation among males and females. These differences ultimately influence the outcomes of antihypertensive therapies and highlight the need for sex-specific treatments in the management of heart failure.

One innovative approach in model-informed drug discovery and development is quantitative systems pharmacology, which supports program decisions from exploratory research through late-stage clinical trials. The studies highlighted here show that, by building a computational platform that integrates the renal and/or cardiac function modulated by the regulatory mechanisms, we end up with a tool for evaluating and understanding the effect of various therapies on cardiac and renal health, including heart failure.

The models discussed in this review are overall simplistic representations of the cardiac and renal physiology that aim to find a balance between accurately representing key physiological behaviors and being simple enough to facilitate integrative dynamic simulations to isolate the consequences of the considered mechanisms. These simplifications may have some effects on the numerical simulations, but they are not expected to impact the overall predicted trends and conclusions. However, various aspects of the cardiac and renal physiology within HF models may need to be refined to more fully represent the pathophysiological processes of heart failure and to thus obtain a more representative model framework.

As discussed, renal impairment is common among HF patients. The association between HF and renal dysfunction is well known; while HF increases the risk of renal insufficiency, CKD increases the risk of hospitalization and mortality [140]. CKD is frequently characterized by a progressive decrease in GFR, and the appearance of albuminuria represents a major risk factor for the development of kidney failure [141]. Given the negative impact of dialysis and kidney transplantation on patient outcomes, it is highly desirable to slow down the progression of CKD and avoid these interventions. This will also ultimately reduce the high costs associated with kidney transplants [141,142]. Medications for HF can have a negative effect on the kidney, so it takes careful monitoring and frequent blood tests, in clinic, to get the balance right. This is how mathematical modeling could help facilitate this close monitoring by predicting the long-term decline or protection of the kidney function all at once for a given drug or combination therapies.

Recently, SGLT2 inhibitors have emerged as a key therapy to prevent the progression of CKD, following the EMPA-KIDNEY trial [141]. In this study, among a wide range of patients with CKD who were at risk for disease progression, empagliflozin therapy was found to lead to a lower risk of the progression of kidney disease or death from cardiovascular causes than a placebo. These findings provide a new avenue for the use of SGLT2 inhibitors in the particular and critical group of patients with CKD. Thus, quantitative and computational modeling will play a significant role in this new direction regarding the personalization of optimal therapeutic schedules in patients with different stages of CKD.

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
