# Peer review of "Understanding the Mechanisms and Treatment of Heart Failure: Quantitative Systems Pharmacology Models with a Focus on SGLT2 Inhibitors and Sex-Specific Differences"

_pharmaceutics, 2023, doi:10.3390/pharmaceutics15031002_

Round 1

Reviewer 1 Report

In this review, this author summarized the pathophysiology of heart failure, treatment of heart failure and an integrated mathematical model of cardiorenal system. In addition to hat this author also describes about the sex-specific differences between male and female. I think present manuscript discussing interesting topics. However, I also thought this manuscript also requires same revision. I commented those to this author.

1) In abstract section, this author described about the sodium-glucose co-transporter 2 inhibitors (SGLT2) without notifying the impact of SGLT2 on present topic. Then the reader might not understand the reason why this author pointed out SGLT2 but not for other treatments.

2)Even this author described as heart failure is a major clinical and public health challenge in abstract section, there is no description about the prevalence of heart failure in introduction section.

3) Since there are many diseases that cause heart failure. And this author thought heart failure is a major clinical condition. Then this author should described about the prevalence of heart failure by focusing on the causes.

4) According to abstract and introduction of present manuscript, this author focusing on sodium-glucose co-transporter 2 inhibitors (SGLT2). Then the description should be focus on SGLT2. There are many cases of heart faire that is not recommended by treatment of SGLT2.

5) About the treatment of heart failure.

 This author described about the treatment of heart failure by showing the potential mechanisms. However, from the perspective of clinical aspects, the outcomes of treatment of heart failure by each treatment strategy should be taken into consideration. Even title of this manuscript is quantitative systems pharmacology modeling, present manuscript just discussing about the mechanisms that relates to heart failure but not discussing the quantitative impact of each treatments. Then previous clinical studies that report the specific outcomes of each treatment should be shown.

6) Even this author focuses on SGLT2, the description about SGLT2 is limited in discussion section.

7) Even this author describes about the general mechanism that relates to heart failure, present conclusion that is described in discussion section emphasize the impact of CKD condition. Then conclusion of present study should be revised.

8) Clinical perspective of present manuscript also should be clarified in discussion section.

9) In discussion section, there is no description about the sex-specific differences in heart failure which is a one of the main topics of present manuscript.

Author Response

Reviewer 1

In this review, this author summarized the pathophysiology of heart failure, treatment of heart failure and an integrated mathematical model of cardiorenal system. In addition to hat this author also describes about the sex-specific differences between male and female. I think present manuscript discussing interesting topics. However, I also thought this manuscript also requires same revision. I commented those to this author.

Thank you for your interest and suggestions.

1) In abstract section, this author described about the sodium-glucose co-transporter 2 inhibitors (SGLT2) without notifying the impact of SGLT2 on present topic. Then the reader might not understand the reason why this author pointed out SGLT2 but not for other treatments.

Thank you for pointing this out, we apologize for the oversight. We made the following changes in the revised Abstract:

“Sodium-glucose co-transporter 2 (SGLT2), a recent antihyperglycemic drug, have been found to significantly improve HF complications and mortality. They act through many pleiotropic effects, and show better improvements compared to others existing pharmacological therapies.”

2)Even this author described as heart failure is a major clinical and public health challenge in abstract section, there is no description about the prevalence of heart failure in introduction section.

As suggested by the reviewer, we added these details to the revised Introduction:

“Affecting more than 64 million people worldwide, heart failure is associated with considerable morbidity and mortality, limited functional capacity, reduced quality of life, and substantial financial burden [2]. It remains a highly prevalent disease among older adults and is associated with a significant risk of mortality within the first year after diagnosis.”

3) Since there are many diseases that cause heart failure. And this author thought heart failure is a major clinical condition. Then this author should described about the prevalence of heart failure by focusing on the causes.

We appreciate the suggestion and believe our previous answer and the following additions in the Introduction responds to the reviewer’s concern:

“HF is caused by structural abnormalities of the heart, functional abnormalities, and other triggering factors [5]. The most common etiologies are ischemic heart disease, hyper-tension, and diabetes [3–5]. Other causes include cardiomyopathies, valvular disease, uncontrolled arrythmias, myocarditis, pericarditis, and toxins (e.g., alcohol, cytotoxic drugs).”

4) According to abstract and introduction of present manuscript, this author focusing on sodium-glucose co-transporter 2 inhibitors (SGLT2). Then the description should be focus on SGLT2. There are many cases of heart faire that is not recommended by treatment of SGLT2.

We agree with the reviewer’s assessment. We made clarifications about the efficacy of SGLT2 inhibitors in patients with reduced ejection fraction in the revised Introduction and in section 3.2 in our revisions:

“However, unlike other antihyperglycemic agents, they have been shown to reduce the risk of hospitalization due to HF-rEF and serious renal outcomes in patients with diabetes [59–61].”

Further, we added to the revised Discussion where we now mention the promising results observed in patients with preserved ejection fraction:

“Therefore, further studies have been conducted to examine the efficacy of these inhibitors in patients with HF-pEF (EMPEROR-Preserved trial [139]). The outcomes of this trial showed encouraging results as empagliflozin reduced the combined risk of hospitalization for HF or cardiovascular death in patients with HF-pEF, regardless of whether they have diabetes. These findings imply that SGLT2 inhibitors may be the first class of medication to enhance cardiovascular outcomes in individuals with HF-pEF.”

5) About the treatment of heart failure.

This author described about the treatment of heart failure by showing the potential mechanisms. However, from the perspective of clinical aspects, the outcomes of treatment of heart failure by each treatment strategy should be taken into consideration. Even title of this manuscript is quantitative systems pharmacology modeling, present manuscript just discussing about the mechanisms that relates to heart failure but not discussing the quantitative impact of each treatments. Then previous clinical studies that report the specific outcomes of each treatment should be shown.

We think this is great suggestion. In response, we revised paragraph 3.1 as follows.

For digitalis:

 “It is commonly believed that digitalis, by increasing cardiac intracellular calcium (Ca^(2+)), provides both inotropic and arrhythmogenic effects [47].”

For diuretics:

“As the compensatory mechanisms of HF involve fluid retention, diuretics were included to relieve pulmonary congestion and peripheral edema [12] by reducing extracellular fluid volume. They act by reducing sodium reabsorption at different sites in the nephron, resulting in an increase in urinary sodium and water losses As the compensatory mechanisms of HF involve fluid retention, diuretics were included to relieve pulmonary congestion and peripheral edema [12] by reducing extracellular fluid volume. They act by reducing sodium reabsorption at different sites in the nephron, resulting in an increase in urinary sodium and water losses.”

For antihypertensive drugs:

The description of their counteraction of the deleterious effects of the RAAS is discussed in section 2.3, where we further describe these mechanisms.

For beta-blockers:

“For example, by reducing the β_1-receptors, beta-blockers alter the effects of catecholamines, epinephrine, and norepinephrine, reducing the heart rate with less contraction, which thus contributes to lower blood pressure and prevent from arrhythmias For example, by reducing the β_1-receptors, beta-blockers alter the effects of catecholamines, epinephrine, and norepinephrine, reducing the heart rate with less contraction, which thus contributes to lower blood pressure and prevent from arrhythmias [54].”

Per the suggestions of this and other reviewers, we also amended the title of our review in our revised manuscript.

6) Even this author focuses on SGLT2, the description about SGLT2 is limited in discussion section.

Thank you to the reviewer for pointing this out. Accordingly, we changed the revised Discussion section as follows:

“SGLT2 inhibitors, a novel class of antihyperglycemic drug, have been surprisingly found to significantly reduce hospitalizations for HF and cardiovascular mortality in patients with HF-rEF, potentially due to their numerous pleiotropic effects. Therefore, further studies have been conducted to examine the efficacy of these inhibitors in patients with HF-pEF (EMPEROR-Preserved trial [139]). The outcomes of this trial showed encouraging results as empagliflozin reduced the combined risk of hospitalization for HF or cardio-vascular death in patients with HF-pEF, regardless of whether they have diabetes.”

7) Even this author describes about the general mechanism that relates to heart failure, present conclusion that is described in discussion section emphasize the impact of CKD condition. Then conclusion of present study should be revised.

We revised the Discussion section in our revisions to respond to this concern:

“HF is a globally prevalent medical condition that arises due to structural defects in the heart that hinder its ability to pump blood efficiently at normal cardiac pressures, thereby failing to meet the body’s metabolic demands. These irregularities are frequently caused by ischemic heart disease, hypertension and diabetes. Therapeutic approaches involve the use of diuretics to remove excess fluids, inhibition of hyperactive neuro-hormonal systems (e.g., antihypertensive medications), and improvement of cardiac contractility (e.g., digitalis, beta-blockers). Currently, there is no evidence-based pharmacological therapy for HF-pEF, possibly due to the type of hypertrophy we have (i.e., concentric hypertrophy), which results in an abnormal filling pattern rather than a volume overload as observed in HF-rEF.”

8) Clinical perspective of present manuscript also should be clarified in discussion section.

We appreciate the suggestion. In response, we added the following discussion of clinical perspectives by mentioning the recent studies that were conducted in CKD patients on page 15 in our revisions:

“Recently, SGLT2 inhibitors have emerged as a key therapy to prevent the progression of CKD, following the EMPA-KIDNEY trial [141]. In this study, among a wide range of patients with CKD who were at risk for disease progression, empagliflozin therapy was found to lead to a lower risk of progression of kidney disease or death from cardiovascular causes than placebo. These findings provide a new avenue for the use of SGLT2 inhibitors in the particular and critical group of patients with CKD. Thus, quantitative and com-putational modeling will play a significant role in this new direction regarding the personalization of optimal therapeutic schedule in patients with different stages of CKD.”

9) In discussion section, there is no description about the sex-specific differences in heart failure which is a one of the main topics of present manuscript.

Thank you for pointing this out. We added to the revised Discussion section to address this:

“Beyond specific treatments, there have also been documented instances of sexual di-morphism in cardiovascular and renal function, leading to disparities in blood pressure regulation among males and females. These differences ultimately influence the outcomes of antihypertensive therapies and highlight the need for sex-specific treatments in the management of heart failure.”

Reviewer 2 Report

This is a comprehensive review of heart failure and CKD and use of SGLT2 inhibitors. This could be reflected in the title and with the QSP reference in the title at the moment I felt more could be made of the modelling approaches perhaps with an additional flow diagram

Author Response

Reviewer 2

Comments and Suggestions for Authors

This is a comprehensive review of heart failure and CKD and use of SGLT2 inhibitors. This could be reflected in the title and with the QSP reference in the title at the moment I felt more could be made of the modelling approaches perhaps with an additional flow diagram.

Thank you to the reviewer for their interest and review of our manuscript. Per their suggestion and those of the other reviewers, we have amended the title to be “Understanding the mechanisms and treatment of heart failure: quantitative systems pharmacology models with a focus on SGLT2 inhibitors and sex-specific differences”. We have opted to not include an additional flow diagram, since every mathematical model will be slightly different. We believe that the figures included in our paper and the references to previous modelling work help guide the reader.

Reviewer 3 Report

In this review, the authors first introduced the pathophysiology of heart failure (HF) and its correlation with RAAS. Then, the author presented both traditional and recent methods to treat HF, including mechanisms behind a novel SGLT2 inhibitor treatment. Lastly, the authors introduced modeling systems to predict renal and cardiovascular functions before hypothesizing that using a QSP model, which integrates renal function, cardiac function, and sex differences, would promote drug discovery and development.

The authors created a thorough introduction to heart failure and its treatments. However, authors should consider including the models for the renal and cardiac functions with their respective parameters (concern 1)—for example, the model from references 86, and 87 with the solute transporter SGLT2. Therefore, the authors can discuss the potential parameters involved in the QSP model and predicting outcomes (concern 2). The other concern (3) is that the paper title indicates the QSP model has been applied in HF therapeutics, while it’s more like a theoretical concept based on the content. The authors should revise the title to not mislead people.

Author Response

Reviewer 3

Comments and Suggestions for Authors

In this review, the authors first introduced the pathophysiology of heart failure (HF) and its correlation with RAAS. Then, the author presented both traditional and recent methods to treat HF, including mechanisms behind a novel SGLT2 inhibitor treatment. Lastly, the authors introduced modeling systems to predict renal and cardiovascular functions before hypothesizing that using a QSP model, which integrates renal function, cardiac function, and sex differences, would promote drug discovery and development.

The authors created a thorough introduction to heart failure and its treatments. However, authors should consider including the models for the renal and cardiac functions with their respective parameters (concern 1)—for example, the model from references 86, and 87 with the solute transporter SGLT2.  Therefore, the authors can discuss the potential parameters involved in the QSP model and predicting outcomes (concern 2).

Thank you for your suggestions. In response to both Concerns 1 and 2 we included the portions of the model related to Figures 1 and 2 in revised Sections 4.1 and 4.2, respectively.

The other concern (3) is that the paper title indicates the QSP model has been applied in HF therapeutics, while it’s more like a theoretical concept based on the content. The authors should revise the title to not mislead people.

We appreciate the suggestion and in response to similar comments from the other reviewers, ee amended the title in our revisions. 

Reviewer 4 Report

This is an interesting review of the subject matter. The manuscript was well written, and the content is relevant to the readership of this journal.

Author Response

Reviewer 4

Comments and Suggestions for Authors

This is an interesting review of the subject matter. The manuscript was well written, and the content is relevant to the readership of this journal.

We thank the reviewer for their interest and enthusiasm about our review.